# ceRNAR: An R package for identification and analysis of ceRNA-miRNA triplets

**Yi-Wen Hsiao**[1], **Lin Wang**[1], **Tzu-Pin Lu**[1,2]*

**1** Institute of Epidemiology and Preventive Medicine, Department of Public Health, College of Public Health, National Taiwan University, Taipei, Taiwan, **2** Bioinformatics and Biostatistics Core, Center of Genomic and Precision Medicine, National Taiwan University, Taipei, Taiwan

☯ These authors contributed equally to this work.

* tplu@ntu.edu.tw

**Data Availability Statement:** The codes are published online in the github (https://github.com/ywhsiao/ceRNAR). The analyzed data (TCGA-LUAD and TCGA-LUSC) are all retrieved from public

## Abstract

Competitive endogenous RNA (ceRNA) represents a novel mechanism of gene regulation that controls several biological and pathological processes. Recently, an increasing number of *in silico* methods have been developed to accelerate the identification of such regulatory events. However, there is still a need for a tool supporting the hypothesis that ceRNA regulatory events only occur at specific miRNA expression levels. To this end, we present an R package, ceRNAR, which allows identification and analysis of ceRNA-miRNA triplets via integration of miRNA and RNA expression data. The ceRNAR package integrates three main steps: (i) identification of ceRNA pairs based on a rank-based correlation between pairs that considers the impact of miRNA and a running sum correlation statistic, (ii) sample clustering based on gene-gene correlation by circular binary segmentation, and (iii) peak merging to identify the most relevant sample patterns. In addition, ceRNAR also provides downstream analyses of identified ceRNA-miRNA triplets, including network analysis, functional annotation, survival analysis, external validation, and integration of different tools. The performance of our proposed approach was validated through simulation studies of different scenarios. Compared with several published tools, ceRNAR was able to identify true ceRNA triplets with high sensitivity, low false-positive rates, and acceptable running time. In real data applications, the ceRNAs common to two lung cancer datasets were identified in both datasets. The bridging miRNA for one of these, the ceRNA for *MAP4K3*, was identified by ceRNAR as hsa-let-7c-5p. Since similar cancer subtypes do share some biological patterns, these results demonstrated that our proposed algorithm was able to identify potential ceRNA targets in real patients. In summary, ceRNAR offers a novel algorithm and a comprehensive pipeline to identify and analyze ceRNA regulation. The package is implemented in R and is available on GitHub (https://github.com/ywhsiao/ceRNAR).

## Author summary

The gene expression regulating mechanisms in humans are complex as many regulators are highly connected and are compensatory to each other. Not to mention, a large

domains (GDG data portal: https://portal.gdc.cancer.gov/).

**Funding:** This study was partly supported by grants from the Ministry of Science and Technology, Taiwan (MOST-106-2314-B-002-134-MY2 (recipient: TPL), MOST-108-2314-B-002-103-MY2 (recipient: TPL), and MOST-109-2314-B-002-151-MY3, recipient: TPL), and the "National Taiwan University Higher Education Sprout Project (NTU-110L8810, recipient: TPL)" within the framework of the Higher Education Sprout Project by the Ministry of Education (MOE) in Taiwan. The funders had no role in study design, data collection and analysis, decision to publish, or preparation of the manuscript.

**Competing interests:** The authors have declared that no competing interests exist.

proportion of the potential interactions between miRNA and gene expression remain unclear due to the challenges and difficulties of performing biological experiments to validate them. With the advancement in high-throughput genomic technologies, massive data of different molecules can be generated within a short period of time. However, utilizing such massive data towards unveiling the regulatory relationships through computational methods and statistical models poses a bottleneck. To address this issue, we present an R package, *ceRNAR*, that enables researchers to explore and identify potential competing endogenous RNA (ceRNA) events through three consecutive steps, and provides novel biological insights into the analytical results. ceRNA constitutes of a set of different RNAs that compete with messenger RNA for interacting with a given miRNA, towards gene expression regulation. Through our proposed tool, users can avail a novel algorithm and a comprehensive pipeline for identifying novel regulators and interactions among miRNA and messenger RNA that may potentially explain biological and pathological processes.

## Introduction

Regulation of gene expression can occur at multiple levels via both transcriptional and post-transcriptional mechanisms [1]. Many non-coding RNAs have critical roles in post-transcriptional regulation of protein-coding genes [2]. MicroRNAs (miRNAs) are short, non-coding, single-stranded RNAs with ~22 nucleotides. They usually bind protein-coding genes via partial complementarity with many miRNA response elements (MREs) to repress gene expression by inhibiting translation. Previous studies have shown that miRNAs are involved in a broad range of cancer-associated biological processes, including apoptosis, proliferation, metastasis, and angiogenesis [3]. Similar to gene expression, miRNA expression has cancer-specific patterns that can be used to detect cancers. Therefore, the expression values of RNA can serve as diagnostic, prognostic, or therapeutic biomarkers in a diverse range of cancers [4].

The concept of competing endogenous RNAs (ceRNAs), also called miRNA sponges or miRNA decoys, has revolutionized our knowledge of miRNA regulatory mechanisms. Such RNAs include canonical protein-coding messenger RNAs (mRNAs), long non-coding RNAs (lncRNAs), circular RNAs (circRNAs), and pseudogenes [5]. Their mechanism is to compete with miRNAs for binding their regulatory sequences. There are two primary hypotheses regarding the regulatory function of ceRNAs, based on their expression level or their number of MREs [6]. Taking miRNA-mRNA regulation for example (i.e., where two mRNAs act as ceRNAs that can bind to the same miRNA), the first hypothesis is that the miRNA tends to be sequestered by the mRNA with the higher expression level, leading to weakened inhibitory effects of the miRNA on the other mRNA and thus increasing the expression of the other mRNA under the assumption of equal MREs on the two RNAs. The second hypothesis is that the miRNA has a greater affinity for the mRNA with more MREs in its sequence.

Some ceRNAs have been identified in multiple cancers; for instance, *PTEN* is an important tumor suppressor gene that was also reported to encode ceRNA in prostate cancer [7], glioblastoma [8], and melanoma [9]. This suggests that elucidating ceRNAs can improve the understanding of biological mechanisms in regulating cancer cells. However, using biological experiments to identify ceRNAs is time-consuming and labor-intensive. To address this issue, an increasing number of computational methods have been developed for identifying ceRNAs.

The traditional algorithm is based on the probability theory that two mRNAs share miRNAs and their binding sites (i.e., MREs) [10, 11]. A hypergeometric test is applied to find out if

the probability of binding of an mRNA to a given miRNA is larger than that which would occur by chance. However, such an approach usually uses a selected threshold to choose significant genes and takes only these genes into consideration; it may not extend to the whole-genome level and fails to consider the correlation among genes because this approach treats each gene independently [12].

Recently, because of the popularity of and advances in genomic sequencing technology, more and more mRNA and miRNA gene expression profiles at the whole-genome level have been released publicly [13–15]. However, the analytical results of such continuous data may tend to be sensitive to the outliers in the sample and to the size of the dataset [16]. Therefore, a second approach has been developed based on the observation of linear correlations between pairs of mRNAs that suggest they have a higher chance of competing with a specific miRNA [17–19]. Unfortunately, such a method ignores the contribution of the expression of the miRNA in a ceRNA binding event. Additionally, it also uses a permutation test to estimate mRNA pairs with significant correlation results; sometimes, it has a higher computational cost. To overcome these drawbacks, in this study, we present a novel rank-based algorithm considering the contribution of miRNA expression in a ceRNA binding event and extending the pairwise correlation approach to identify ceRNA-miRNA triplets. All the steps in this algorithm have been incorporated into a user-friendly R package called ceRNAR, which also provides several downstream analyses to further interpret the biological meaning of identified ceRNA events for its users.

## Results

### Simulation results

To evaluate the performance of our proposed method, simulations of mRNA and miRNA expression data in 100 samples were performed in different scenarios (S1 Table). Notably, we focused on the sensitivity and the positive predictive value (PPV) because the number of ceRNA triplets was small among all possible combinations of triplets.

In the beginning, we presumed the sample distribution of each gene follows normal distribution because about 98% of genes' sample distributions passed the normality test based on 9,835 samples in The Cancer Genome Atlas (TCGA) pan-cancer atlas (S1 Fig). Therefore, we firstly presumed synthetic expression data were generated from a multivariate normal distribution with a mean value of 0 and a covariance matrix whose entries are 0.9. This ensured that the ceRNAR algorithm supports the hypothesis that pairs of ceRNA binding partners of a specific miRNA are highly correlated and that it works well to sensitively identify them among a pool of target pairs. However, it is unclear whether such an event between two target genes would occur at the lower or higher expression of a specific miRNA. Therefore, five scenarios were designed to capture how the molecular elements within a triplet interact with each other, and simulations of different parameters were performed. Notably, the number of identified ceRNAs dropped as the window size increased (S2 and S3 Tables, Fig 1). This is because more uncorrelated samples were included in the analysis when longer window sizes were used, especially larger than 30%. In other words, higher noise was included in the analysis, resulting in difficulty in identifying true ceRNA triplets. Next, we simulated different scenarios in which the ceRNA peak was located at different miRNA expression levels (scenarios 1 to 4). As shown in S2 Table and Fig 1A, the performance of our proposed algorithm was highly similar across the four scenarios with true ceRNA signals. Such performance also was shown when removing the permutation test (Fig 1B and S3 Table). Lastly, to reduce the calculation complexity and computation time, we evaluated the performance of the algorithm without the random walk step, which we called the "fast" version. Only minor differences were observed in these two

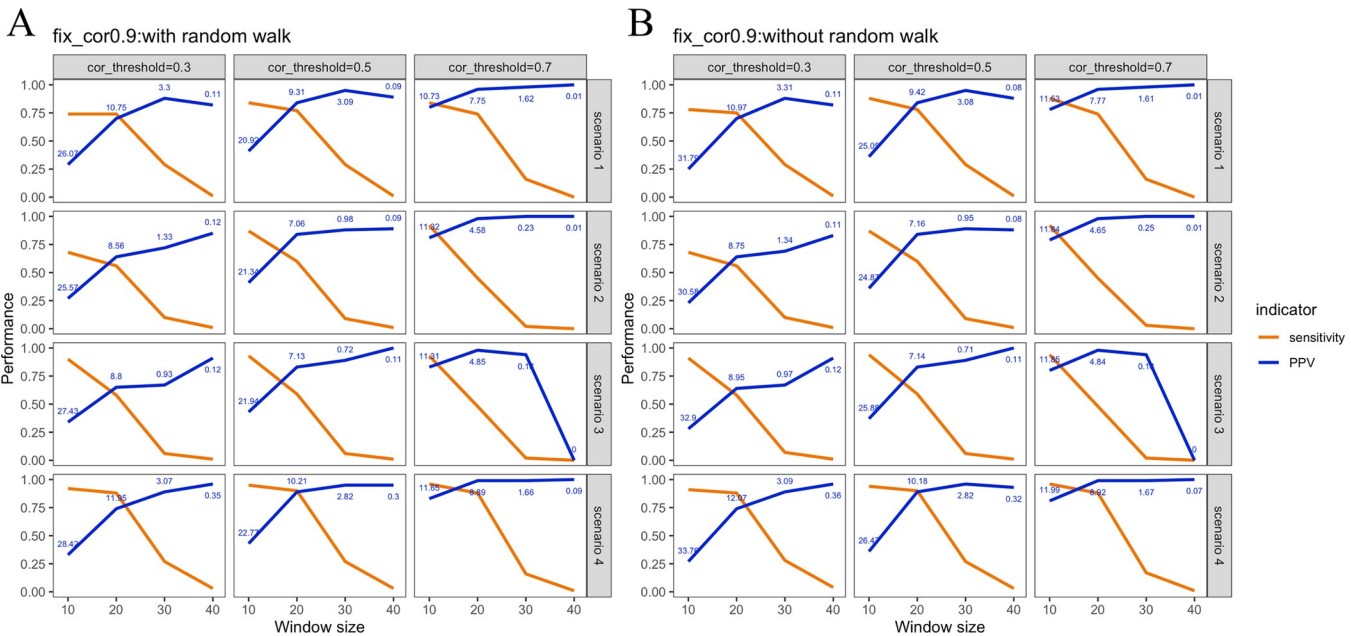

**Fig 1. Performance of our proposed method in four scenarios (1 to 4).** (A) The complete version. (B) The fast version. The numbers in blue represent the average number of identified ceRNA-miRNA triplets after 100 simulations.

versions (0.1 to 0.25 in terms of PPV value, S2 and S3 Tables), suggesting that both versions were able to identify true ceRNA triplets without reporting high false positive results (Fig 1).

In addition to the analytical parameters for the algorithm, a simulation of different proportions of the correlated samples was performed. Because no major differences were observed in the simulated scenarios and running versions, only scenario 3 was evaluated in this round of testing. As shown in S4 Table, four settings for the proportion of correlated samples were analyzed (10%, 20%, 30%, and 40%). Notably, the proposed algorithm showed similar performance in the three highest settings (Fig 2A).

To further examine whether more false positive ceRNA triplets are reported in the fast version, a null scenario without a true ceRNA signal was simulated (scenario 5). The results showed that the negative predictive values of these versions were all higher than 0.9 (S1 and S2 Tables), suggesting there is a low chance of identifying false positive signals in both versions of the algorithm (Fig 2B). Subsequently, we examined how much time can be saved by using the fast version of the proposed algorithm, which omitted the random walk step. On average, the fast version accelerated the algorithm by approximately 76 times (283.967 seconds versus 21,600 seconds), and only slightly higher false positive rates were reported (Fig 2B). Lastly, to evaluate whether different window sizes for merging peaks are critical to the performance of the proposed algorithm, four settings of the proportion of correlated samples and merging window sizes were analyzed. As shown in Fig 2C and S5 Table, the performance of the proposed algorithm was not sensitive to the window sizes for merging peaks.

In addition, to ensure the algorithm only sensitively identified ceRNA events under higher correlation values between them, we also generated simulated data from a multivariate normal distribution with the same mean value but a covariance matrix whose entries are 0.6 (S2A Fig and S6 Table) or 0.3 (S2B and S7 Table). Although the ceRNAR algorithm was still able to detect ceRNA pairs with moderate sensitivity values (0.4 to 0.5) when the correlation values between target genes within a pair were 0.6, it did not work well (sensitivity values were all below 0.25) when the correlation values between target genes within a pair were 0.3. Together,

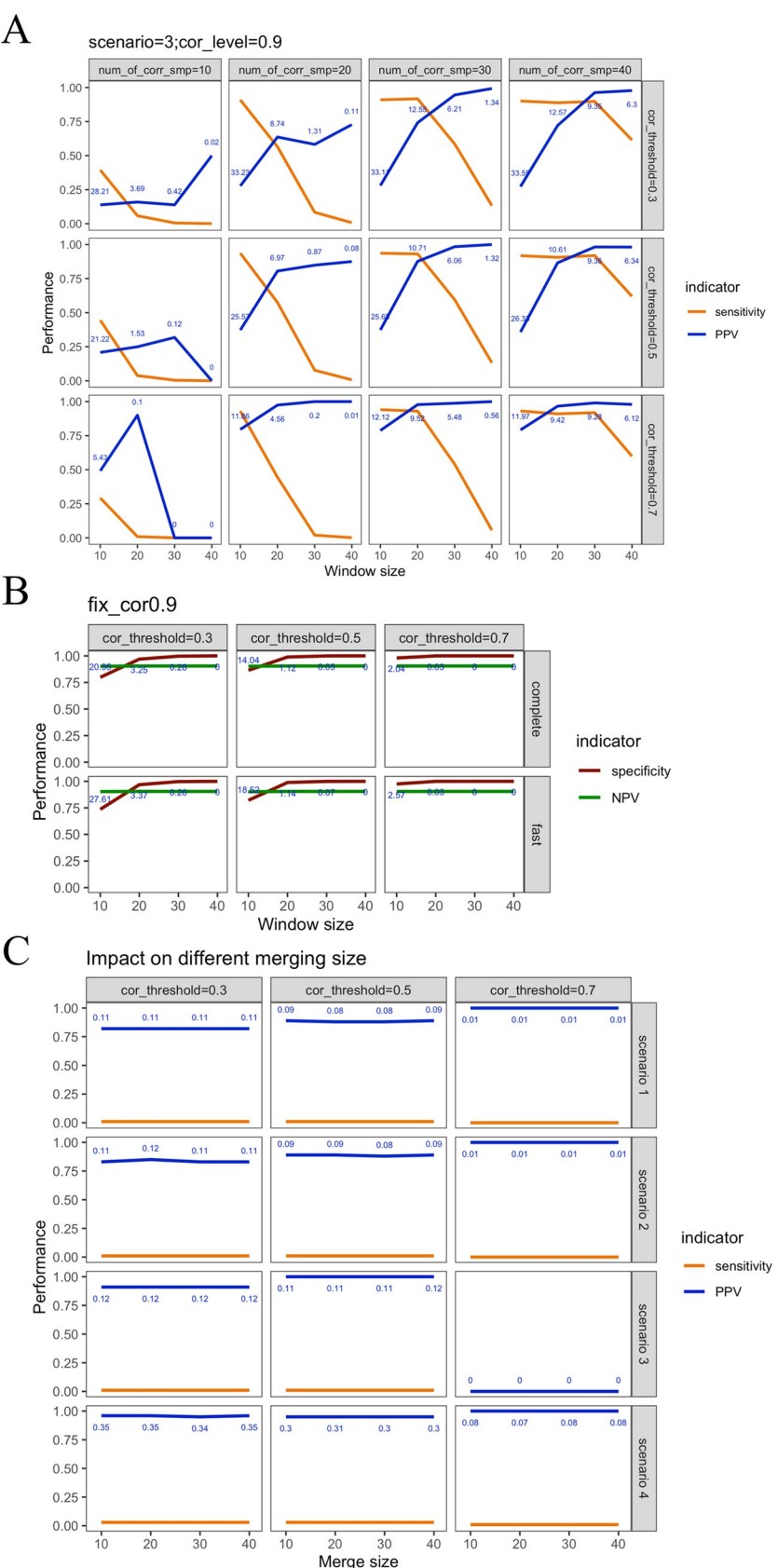

**Fig 2. Performance of our proposed method in three extended cases.** (A) Different proportions of correlated samples in scenario 3 using the fast version. (B) Comparison between complete and fast versions in the null scenario (i.e., scenario 5). (C) Different distances for merging peaks when window size is equal to 40 using the complete version. The numbers in blue represent the number of identified ceRNA-miRNA triplets.

these results support our correlation-based hypothesis that ceRNA events tend to occur when they are highly correlated. However, the above-mentioned results were based on simple and naïve simulations. We also mimicked the distribution of expression from pan-cancer data (9,835 samples), allowing the synthetic data to be generated from a multivariate normal distribution with a randomly selected mean value and a covariance matrix whose entries are also randomly selected. The simulation studies were also performed under different correlation levels. As shown in S3 Fig and S8–S11 Tables, the performance of the ceRNAR algorithm was determined by the level of the correlation values. Nevertheless, it still performed well (sensitivity values > 0.75) when the correlation between genes was high and when the window size was 10 throughout all scenarios (S3B Fig.), suggesting the ceRNAR algorithm is efficient to identify most potential ceRNA pairs when their correlation pattern is relatively high (0.8–0.9) to compete with a specific miRNA, and such a pattern exists in at least 20% of the sample. The optimal parameter settings for real data were also observed. For 100 samples, the best window size was 10, and the best cutoff correlation value for selecting the most significant event was 0.7.

## Comparison with other tools

We have compared our tool with other state-of-the-art tools, including SPONGE, JAMI, GDCRNATools, and CERNIA, in terms of their performances using synthetic data and their running time using real data. Fig 3A and S12 Table illustrate that ceRNAR workflow generally outperformed the other tools in terms of sensitivity and PPV in all scenarios when the window size is set to 10 and the correlation cutoff is set to 0.7. It can be mainly observed when the correlation values among correlated genes are relatively high. However, all of the tools could identify valid ceRNA triplets without reporting high false-positive results except JAMI (Fig 3B and S12 Table). GDCRNATools generally had a high sensitivity compared to the other tools, and a lower specificity than ceRNAR, suggesting that it is good for catching ceRNAs but comes with a relatively high rate of false positives. Regarding running time, we used different sample numbers (250, 500, and 100) on a small subset of the pan-cancer dataset with 15 genes which form 105 triplets; we also used different triplet numbers (105, 1,225, and 4,950) on a small subset of the pan-cancer dataset with 250 samples (S4 Fig and S13 Table). Although the ceRNA algorithm was not fast with a large sample size and a large number of triplets compared with SPONGE, GDCRNATools and CERNIA, it was slightly faster than JAMI.

## Application to TCGA cancer cohort datasets

To further validate the applicability and robustness of the ceRNAR algorithm, we also applied the algorithm to two TCGA-derived lung cancer cohorts–TCGA-LUAD and TCGA-LUSC (S14 Table). The top bridging miRNAs and the hub genes among ceRNA triplets are shown in S15 and S16 Tables. Intriguingly, the two cancer cohorts (LUAD and LUSC) shared some common triplets involving 53 miRNAs and 905 ceRNAs, which allowed construction of a miRNA-modulated ceRNA regulatory network (S5 Fig). Among them, *PLEKHG6* had the largest number of co-expressed ceRNAs, and of the 53 common miRNAs, the top three miRNAs that bridged over 20 ceRNA pairs were hsa-miR-183-5p, hsa-miR-133a-3p, and hsa-miR-142-5p. *MAP4K3* was another common hub gene in both datasets, and its bridging miRNA was hsa-let-7c-5p, around which a regulatory network of corresponding ceRNAs was built

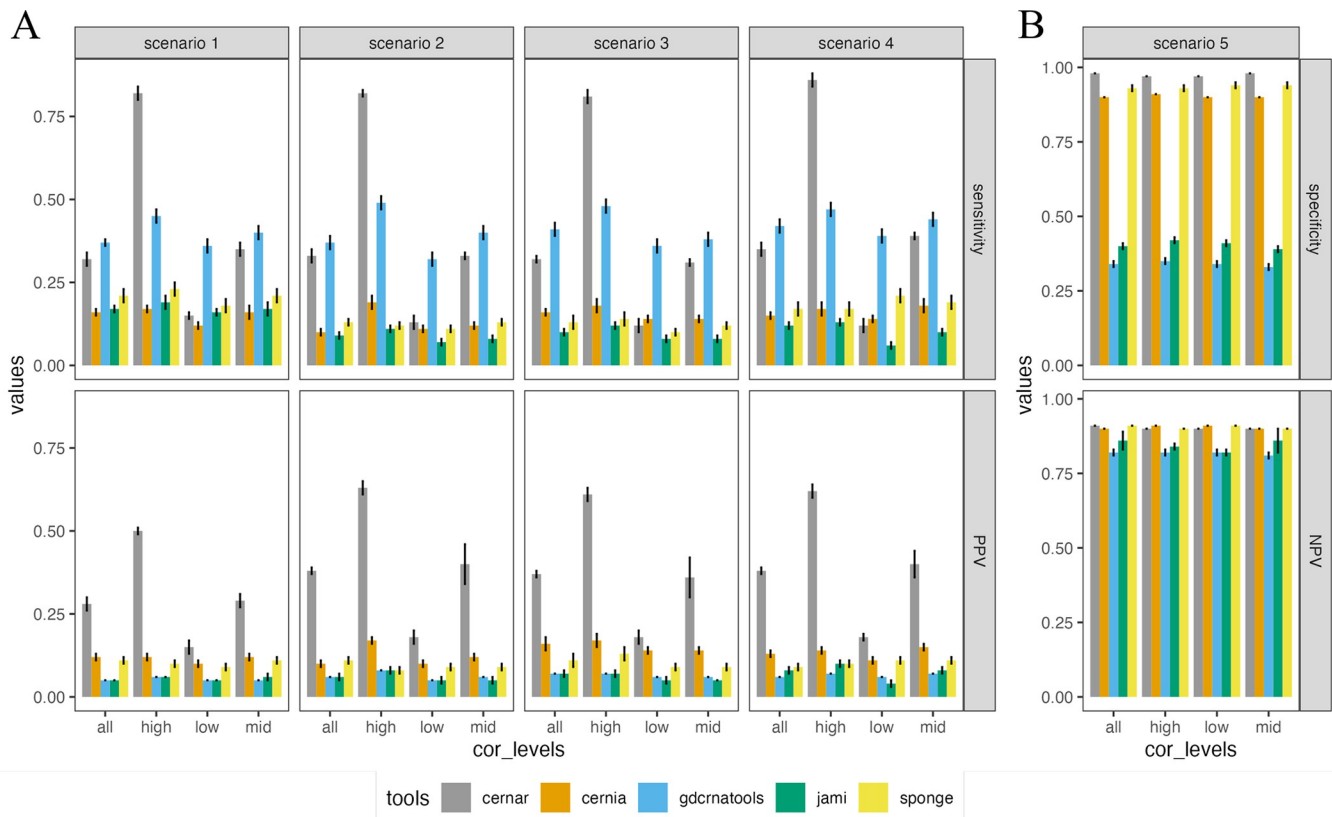

**Fig 3. Performance comparisons between ceRNIA, JAMI, SPONGE, GDCRNATools and ceRNAR packages using synthetic data that mimic the real-world case.** (A) Sensitivity and PPV of various tools at different correlation levels for 100 samples of 105 pairs of target genes under four scenarios (1 to 4). (B) Specificity and NPV of various tools at different correlation levels for 100 sample of 105 pairs of target genes under null scenario (scenario 5). "cor_levels" represents four different correlation levels (all: 0.3–0.9; high: 0.8–0.9; low: 0.3–0.4; mid: 0.5–0.7).

(S6A Fig), and the expression level related to the regulatory occurrence of its bridging ceRNAs is displayed in S6B Fig. Since lung cancers share some common molecular characteristics, such results demonstrate the applicability and robustness of the ceRNAR algorithm in multiple cancers or diseases.

In addition, we compared our findings against experimentally validated miRNA-gene pairs in the miRSponge database to endorse the potential ceRNAs identified by ceRNAR. As shown in S7 Fig, around 1% (ceRNA pairs that can be validated among total ceRNA founded based on ceRNAR algorithm) of experimentally validated ceRNA triplets were identified in LUAD and LUSC. The low proportion may be attributed to the fact that only 158 ceRNA triplets were analyzed and that those ceRNA triplets may not be expressed in lung tissues. Furthermore, approximately 13–14% of ceRNA triplets with at least one experimentally validated miRNA-target interaction were identified by using the ceRNAR algorithm. A Chi-square test was performed to examine whether the findings from the ceRNAR package were significantly enriched in the TCGA data, and the results showed that the P-values obtained from the LUAD and LUSC datasets were both less than 2.2e-16, suggesting our ceRNAR package can successfully identify previously reported ceRNA triplets.

## Discussion

ceRNA events are a newly discovered type of post-transcriptional regulation, and the identification of ceRNA-miRNA triplets using *in silico* methods is an emerging research area.

Therefore, we developed a novel computational algorithm to explore such regulatory events for further biomedical interpretation and application. Our proposed method is based on a simple pairwise correlation approach that considers the miRNA-modulated ceRNA interaction. First, we ranked samples based on their miRNA expression value to include the contribution of miRNA expression and identify which miRNA expression intervals tend to have a higher correlation with pairs of mRNA targets. Secondly, we used a sliding window approach to form more correlation values in a triplet to improve the performance and outcomes of the subsequent statistical approach. Lastly, we applied a cumulation-like approach to sum up the slight changes in correlation values across samples. We used segment clustering to understand the sample clustering in terms of the gene-gene correlations and the miRNA expression intervals so that we could also use sample proportion to support our findings. Several simulations have been conducted for the optimization of the parameters subject to specific ranges of settings, and the robustness of our approach when it does not involve a permutation test has also been evaluated through a simulation study. Connecting with six downstream analyses, our R package may assist researchers to have a deeper understanding of the disease-specific biological regulation and prognostic application for each identified ceRNA-miRNA triplet.

Recently, more and more tools have been developed to identify potential ceRNA events. It is important to systematically evaluate the ceRNAR package in comparison with the five other published ceRNA prediction tools—SPONGE [20], CERNIA [21], GDCRNATools [22], JAMI [23], and CUPID [24]—that are expression-based (rather than sequence-based, i.e., spongeScan [25]). We have compared them in terms of several features, such as miRNA-target data sources, study design, ceRNA classes, ceRNA prediction algorithm, and language for implementation (S17 Table). Noting that JAMI is the multi-threading version of CUPID, we decided to keep only JAMI for further analyses. Thus, we used four algorithms, including SPONGE, CERNIA, GDCRNATools, and JAMI, for the comparisons. Notably, the four algorithms and our ceRNAR all used a similar strategy, which is utilizing miRNA-target data sources to identify potential miRNA-gene/lncRNA/pseudogene pairs from other databases. All four of these algorithms are implemented in R. JAMI is based on conditional mutual information, which is particularly useful to capture non-linear associations by estimating the effect of a miRNA on its target pairs through a permutation test [26]. Excepting JAMI, the rest algorithms consider the correlations between miRNA-mediated genes/lncRNA. Although the majority of such algorithms are correlation-based, some differences still exist. For examples, GDCRNATools is based on sensitivity correlation computation through effectively estimating covariate matrices and also considering the impact of a miRNA on its target pairs. SPONGE also uses sensitivity correlation to quantify the impact of a miRNA on its target pairs (i.e., linear partial correlation), but further applies a null model-based p-value computation to estimate potential ceRNA pairs. It is worth mentioning that CERNIA and JAMI consider both MRE- and expression-based data, whereas SPONGE, GDCRNATools and ceRNAR only analyze genome-wide expression data. Since several studies [18, 27, 28] indicate that ceRNA triplets may be observed in a specific range of miRNA expression, such an approach can help to focus on the true positive region with high signal-to-noise ratio instead of missing the ceRNA triplets due to signal dilution by global noise. Our ceRNAR algorithm showed the highest sensitivity in identifying potential ceRNA triplets (Fig 3).

Regarding the two online servers, miRTissue_ce [29] and Encori (i.e., starBase v2) [30] are two web servers that integrate ceRNA data sources, ceRNA prediction algorithms, and even some data analyses and visualizations that can be easily accessed by the users. Fiannaca *et al.* [29] have compared miRTissue_ce and Encori in terms of many features. Here, we further compared these web servers with our ceRNAR based on these features to see whether there is any add-on value that ceRNAR can provide these web services. First, the interactions between

miRNAs and target genes in ceRNAR are supported by nine databases, including two experimentally validated miRNA-target databases and seven computationally predicted miRNA-target databases. But Encori and miRTissue_ce are supported by 4 and 8 computationally predicted miRNA-target databases, respectively. Second, Encori uses a hypergeometric test to predict ceRNA, and miRTissue_ce integrates that method with a global test and SPONGE. However, one major disadvantage of the hypergeometric test is that the test requires a predefined p-value threshold to select significant genes. When using differentially expressed genes, such an approach is not suitable to be applied to the whole genome, because the hypergeometric test fails to consider the interactions among genes due to its independence assumption about genes. This is why we present a novel rank-based algorithm considering the contribution of miRNA expression in a ceRNA event and extend the pair-wise correlation approach to identify ceRNA-miRNA triplets using whole genomic information. Lastly, these two web servers can predict many types of ceRNA events, but ceRNAR only focuses on one ceRNA event class (i.e., mRNA-miRNA).

In real application, we utilized non-small cell lung cancer (NSCLC) data to evaluate the applicability of ceRNAR. NSCLC accounts for 85% of lung cancer and is one of the most common malignant tumors worldwide [26]. Although there has been progress in successful treatment for NSCLC patients these past several decades, the 5-year survival rate for NSCLC is still relatively low (25%) [31]. Also, the molecular networks involved in NSCLC remain incompletely described in terms of their roles in etiology, progression, and metastasis. Hence, we applied the ceRNAR algorithm to two NSCLC-related cancer datasets in TCGA lung cancer cohorts. Several common miRNAs and ceRNAs identified by the ceRNAR algorithm have also been previously reported by other studies. For example, hsa-miR-183-5p was found to inversely regulate *PTPN4*, serving as a therapeutic target to suppress the metastatic potential in NSCLC patients [32], and hsa-let-7c-5p was verified to prevent cancer metastasis by degrading its bridging hub ceRNA, *MAP4K3* [33]. Although our simulation results suggest that the majority of performance indicators have only slight differences in the four scenarios using both complete and fast versions, and the best parameter setting for window size is 10 and for peak threshold is 0.7, and the fine-tuning of appropriate parameters for non-TCGA datasets still needs to be tested. Nevertheless, these results still demonstrated our proposed method is robust and potentially applicable, allowing it to be extended to studies of other diseases.

However, some limitations still existed in our study. First, a smaller sample size of cancer cohorts (i.e., a smaller window size in our case) may lead to less statistical power of the findings. Second, we presumed a linear relationship between the two ceRNAs in each triplet, but in reality, they were not always linearly correlated. Although we have implemented a sliding window approach to capture such relationships, other methods such as mutual information [34] can also be applied. Moreover, the accuracy of the miRNA target prediction databases we used may have affected the definition of putative ceRNA-miRNA triplets and the outcomes of the ceRNAR algorithm because the mechanisms of some miRNA targeting systems have not been fully understood. It is also worth mentioning that our simulation results were based on a predefined covariance matrix. That is the true positive events were from the correlation-based approaches, and thus such events only showed linear relationships among the elements. Notably, such design may lead to the poor performance of JAMI because their algorithm was developed by using the mutual information strategy, which was able to capture non-linear relationships among ceRNA pairs in addition to the linear ones. Lastly, the majority of our findings from the real case study were novel compared to the miRSponge database, although some of the miRNA-targets contained an interaction that was previously experimentally validated. Perhaps further experimental validation of those triplets that contained one experimentally validated miRNA-target interaction should be prioritized to increase the robustness of

our algorithm and the reliability of the novel findings. Therefore, the consideration of all types of miRNA sponges, the amount of MREs, multiple miRNAs that may compete for the same pair of target genes, nontrivial correlations which involve the comparison of pairwise correlation and pairwise partial correlation, and the minimization of computational time are important key areas for further optimization and extension of the ceRNAR algorithm.

In summary, ceRNAR is a promising tool for the recognition of ceRNA-miRNA triplets and ceRNA-ceRNA interaction networks in many human diseases, and hence will speed up our knowledge of the regulatory mechanisms and functions of ceRNA-miRNA triplets in the pathogenesis of disease, including cancers.

## Materials and methods

### Pipeline of ceRNAR

The ceRNAR package is written in R (version 4.0.5) and is available in the Github repository. The main pipeline of ceRNAR is illustrated in Fig 4 and contains three major components for the identification and analysis of ceRNA-miRNA triplets:

- Data preprocessing

- Identification of ceRNA-miRNA triplets

- Downstream analyses

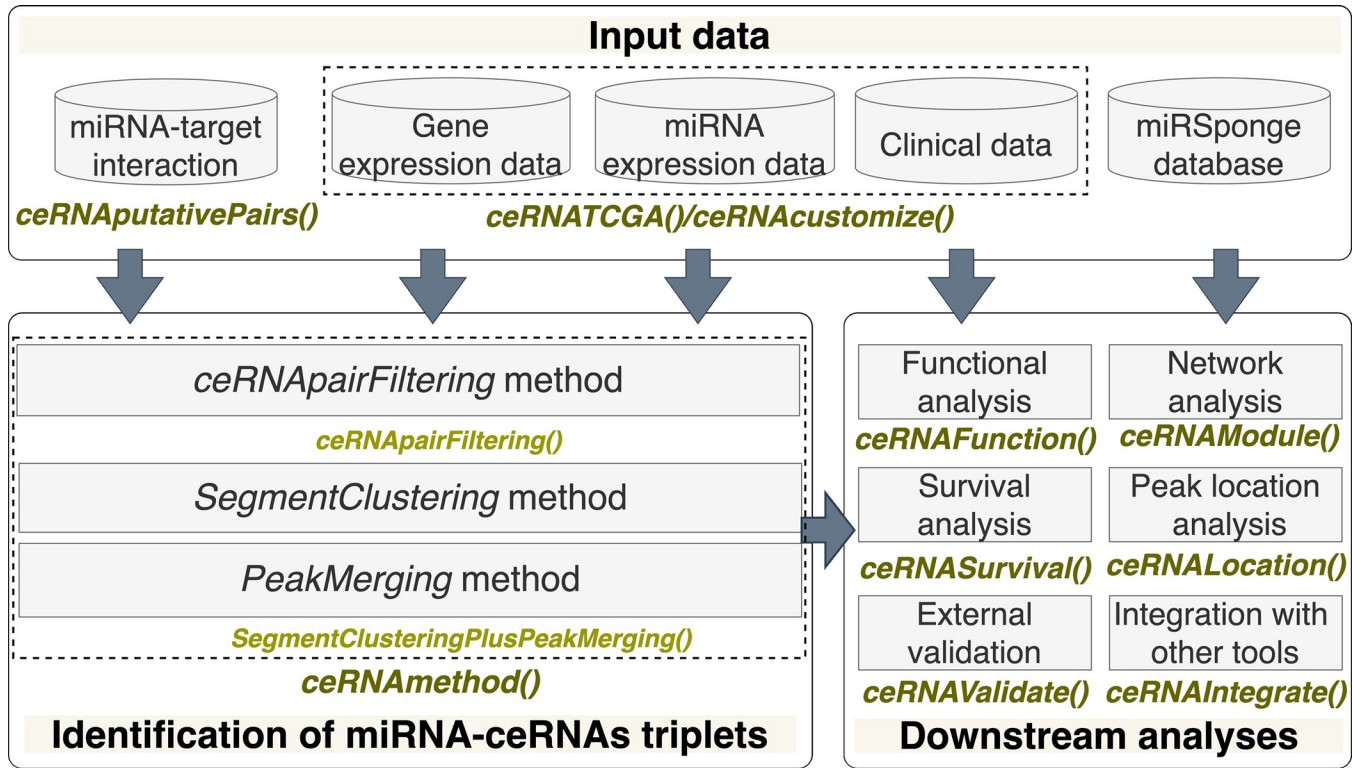

**Fig 4. An overview of our ceRNAR package.** This package includes three main parts. First, "Input data" has three functions: *ceRNAputativePairs()* for extracting curated miRNA-target lists, *ceRNATCGA()* for retrieving TCGA data, and *ceRNAcustomize()* for importing customized data. Second, "Identification of miRNA-ceRNA triplets" involves *ceRNApairFiltering()* and *SegmentClusteringPlusPeakMerging()* and is wrapped into a *ceRNAmethod()* function. Lastly, "Downstream analyses" contains six functions for different tasks, that is, *ceRNAFunction()*, *ceRNAModule()*, *ceRNASurvival()*, *ceRNALocation()*, *ceRNAValidate() and ceRNAIntegrate()*.

To reduce the computational complexity and time cost, the interactions between miRNAs and target genes were based on two experimentally validated miRNA-target databases (miR-TarBase [35] and miRecords [36]) and seven computationally predicted miRNA-target databases (DIANA-micro T-CDS [37], EIMMO [38], miRDB [39], miRanda [40], PITA [41], RNA22 [42] and TargetScan [43]). In the default settings, only those interactions that were validated by experiments and/or predicted by more than half of the databases are retained as target miRNAs and target genes (S8A Fig). Conceptually, the ceRNAR algorithm iteratively goes over each miRNA-target list and runs through each mRNA pair in a list to evaluate the chance of the potential ceRNA event involved. For a specific triplet (i.e., a miRNA and its two targets), their expression vectors are extracted from the original expression matrix. Therefore, a miRNA expression vector, $miRNA_m = [m_{m1}, m_{m2}, \ldots]$, and two mRNA expression vectors, $mRNA_i = [gene_{i1}, gene_{i2}, \ldots]$ and $mRNA_j = [gene_{j1}, gene_{j2}, \ldots]$, are used as inputs into the ceRNAR algorithm to iteratively evaluate whether each mRNA pair is a potential ceRNA event (S8B Fig).

## Data preprocessing

To prepare expression data for further analyses, ceRNAR can automatically retrieve TCGA data, including mRNA expression, miRNA expression, and survival data, by entering the cancer acronym [44], but it also supports the use of customized miRNA and mRNA expression matrices that are pre-normalized and formatted according to the instructions. In ceRNAR, we implement two functions to fulfill these two approaches: *ceRNATCGA* and *ceRNACustomize*.

## Identification of ceRNA-miRNA triplets

To identify miRNA-ceRNA triplets (defined here as a miRNA and two target genes) from expression profiles at a specific miRNA expression level, the *ceRNAMethod* function can be used, and it contains three modules sequentially: *ceRNApairFiltering*, *SegmentClustering*, and *PeakMerging* (Fig 5). We have two assumptions in this study: (1) the expression levels of two target genes tend to be highly correlated when a possible ceRNA event occurs; (2) such events between target genes of a certain miRNA occur at a specific expression interval of that miRNA. However, for each ceRNA triplet from the real data, it is difficult to know which levels of miRNA expression (low, middle, or high expression intervals) will lead to a high correlation between a pair of target genes. Notably, for one miRNA, the high correlation values between two target genes can only be observed in a specific range of the miRNA expression. Definitely, the expression level of one miRNA can be regulated by many factors, such as compensation and/or other interactors. However, with our current understanding of all miRNAs, it is not feasible to consider all potential regulators of one specific miRNA at the same time. Therefore, for one single miRNA, we adapted the approach of utilizing its expression level as the final output instead of considering all possible confounding factors, which can be regarded as the hidden layers of the miRNA expression value.

## The ceRNApairFiltering method

We adopted the sliding window approach to identify the correlation patterns of the two target genes within a specific range of expression values for the miRNA. That is the reason why we ranked the samples based on their miRNA expression value from each triplet to identify the correlation patterns and provided the number of samples that meet the criteria. Therefore, the purpose of this function is to identify ceRNA-miRNA triplets based on the Pearson correlation coefficient through the sliding window approach (S9 Fig) and a running sum statistic for such

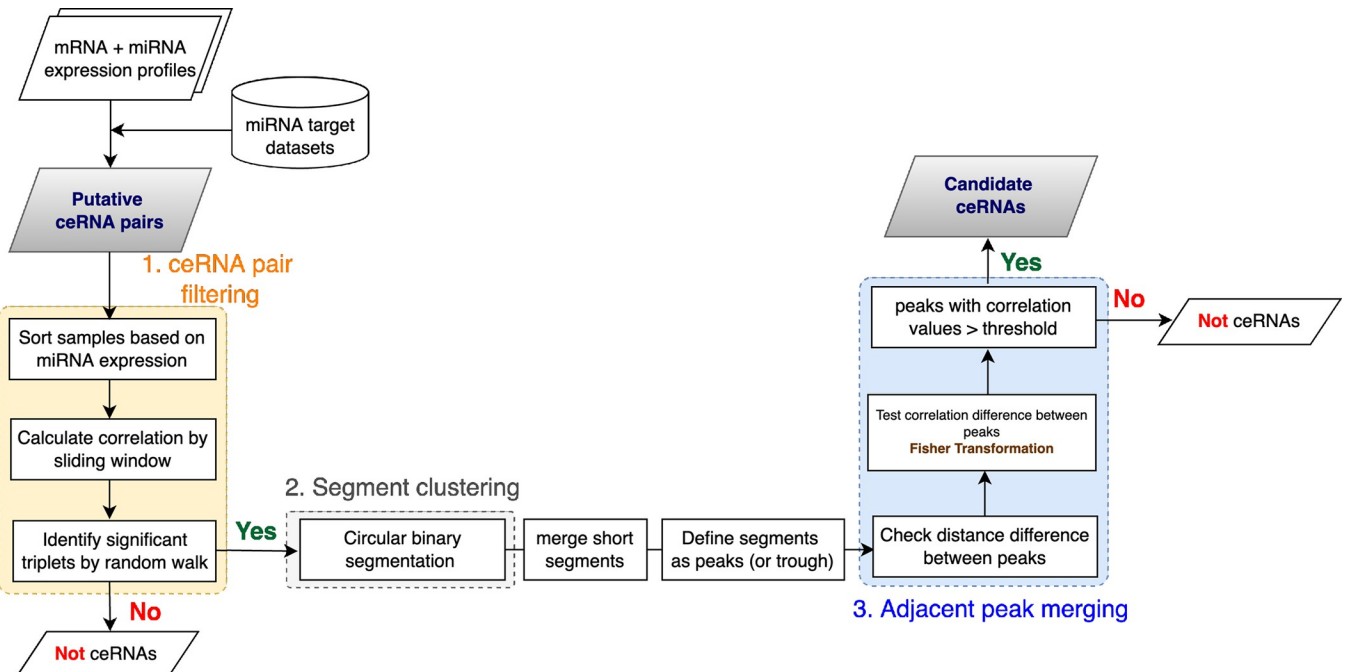

**Fig 5. Three main modules involved in the identification of ceRNA-miRNA triplets.** First, 'ceRNA pair filtering' contains sample sorting based on a miRNA expression vector, data augmentation in terms of correlation calculation through a sliding window, and permutation test by random walk. Next, 'Segment clustering' is based on circular binary segmentation to identify certain miRNA expression intervals (i.e., segments) with the highest correlation between potential ceRNA pairs. Short segments defined by our criteria are further merged and whether a segment is a peak or a trough is defined. Lastly, 'Adjacent peak merging' checks whether a triplet is a candidate ceRNA through two filtering conditions.

values by the random walk approach (S10 Fig). First, the samples are sorted based on their miRNA expression levels. For a putative triplet (gene$_i$, miRNA$_m$, gene$_j$) among N samples, the correlation coefficients of gene expression values between mRNA and miRNA are calculated within each window (i.e., a length of sample size that is always less than N) with predefined window size (w) as follows:

$$r_k^{m,i,j} = corr([gene_k^i, gene_{k+1}^i, \ldots, gene_{k+(w-1)}^i], [gene_k^j, gene_{k+1}^j, \ldots, gene_{k+(w-1)}^j]) \qquad (1)$$

Here k is the number of windows and a predefined integer. Technically, the correlation value between two genes is calculated using the gene expression values of all samples. By using a sliding window approach [45], we can artificially create different varieties of a real dataset to increase its size, which is a sort of data augmentation technique [46].

Because the accumulated changes in terms of correlation values between two genes among samples may tend to increase the chance that a gene pair is highly correlated and, further, will affect a specific phenotype, we borrowed the concept of gene set enrichment analysis (GSEA) [47], which captures the accumulated changes in the expression of all genes within a pathway through a random-walk method, to identify a significant triplet according to the number of the samples enriched in such an event. The main idea of the GSEA algorithm is to understand whether the differentially expressed genes are significantly enriched in the samples belonging to the same phenotype (case or control). First, the differentially expressed genes were ranked by using the differences in expression level between the two phenotypes. Next, the GSEA algorithm gave a positive score to the genes located in the pathway, whereas a negative score was assigned to the genes outside of the pathway. One possible scenario to obtain a high score is that the differentially expressed genes were clustered and enriched in one phenotype instead of

being randomly distributed. This scenario is the same as what we want to identify for the ceRNA triplet. That is, the highest correlation values were observed in a specific range of the miRNA expression and thus we used the two statistics, $P_{obey}(k)$ and $P_{violate}(k)$ to identify the range (S11 Fig). Conceptually, to calculate a score (S) to represent the enriched correlation levels among samples against a specific phenotype, we first rank ordered the k windows to form $L = \{miRNA_1, miRNA_2, \ldots, miRNA_k\}$ based on the average miRNA expression within each window:

$$miRNA_k = \sum_{n=0}^{w-1} m_{k+n}^{m,i,j} \bigg/ w \qquad (2)$$

Next, the S is computed by walking down the window list to evaluated $P_{obey}(k)$, which is the proportion of samples whose gene correlations are over 0.3 (i.e., $r_k^{m,i,j} > 0.3$), weighted by their corresponding correlation and $P_{violate}(k)$, which is the proportion of the rest of the samples at a given ranking window position k across all samples. The formulas are as follows:

$$P_{obey}(k) = \sum_{\substack{r_k \in condition \\ l \leq k}} \frac{|r_l|}{N_R}, \; where \; N_R = \sum_{r_k \in condition} |r_l| \qquad (3)$$

$$P_{violate}(k) = \sum_{\substack{r_k \notin condition \\ l \leq k}} \frac{1}{N_{violate}}, \; where \; N_{violate} = \sum_{r_k \notin condition} I(r_k \notin condition) \qquad (4)$$

S is then defined as the maximum distance from zero of the substations of $P_{obey}(k)$ from $P_{violate}(k)$. If the samples with similar correlation values are not enriched at a particular miRNA expression interval, it means those gene pairs are not biologically relevant to compete with miRNA in a miRNA-gene triplet; that is, there is no ceRNA event observed in this triplet (S11 Fig). Finally, we accessed the significance of an observed S by comparing it with the theoretical S computed by randomly permuting the expression of the candidate $miRNA_m$ 1,000 times to provide assessment of significance. When an entire sample of potential triplets is evaluated, these p-values will be adjusted by the false discovery rate, which provides the estimated probability of whether a triplet within an entire set of potential triplets is a false positive finding or not. After that, we report the ceRNA pairs with statistical significance of the observed S (e.g., adjusted p-value <0.05) as significant ceRNA interactions. For one miRNA, a low score S will be observed if the two target genes have no correlation within a specific range of miRNA expression. On the other hand, the score S will be high if a large proportion of the samples showed high correlation among the two target genes ($P_{obey}(k)$ is high). Consequently, we can use the score S to identify a ceRNA triplet when S is high, and a statistical approach was performed to ensure that S cannot be identified randomly.

## The SegmentClustering method

The motivation of the segment clustering is to group the samples showing high correlation between the expression levels of the two target genes into one single cluster. In our analysis, we divided the samples into small groups (i.e., a window) to calculate the correlation values of the two target genes. Accordingly, we may identify several different groups showing high correlations that actually can be clustered into one single cluster because of their similar correlation values. Therefore, in this method, the concept of a circular binary segment algorithm, which was originally designed for change-point problems such as the identification of copy number

variation, is used [48] to evaluate whether those small windows showing high correlation values should be clustered into a larger group. This method has been widely used in the analysis of copy number variations (CNVs) by many algorithms [49, 50]. Since several previous studies [18, 27, 28] indicated that ceRNA triplets may be observed in a specific range, the purpose of such an approach is to explore the clustering patterns of samples in terms of their gene-gene correlation values per triplet and then group samples with similar correlation values so that a certain miRNA expression interval with the highest correlation within a ceRNA pair can be observed. This algorithm starts with all segments (i.e., several intervals of rank-ordered miRNA expression) identified from the whole dataset. Similar to the previous method, the samples are sorted by the miRNA expression values. A recursive test for the change-points is calculated based on the correlation values of each gene-miRNA pair between each set of two neighboring regions of rank-ordered miRNA expression, and it stops when no significant changes can be found in any two segments. The maximal t-statistic with a permutation reference distribution is chosen to obtain the corresponding p-value. Let $X_1, \ldots, X_n$ be the correlation coefficients of two genes, which are indexed by the corresponding miRNA expression of the n samples being studied. The test statistic is given by $Z_c = max_{1 \leq i < j \leq n} |Z_{ij}|$, where $Z_{ij}$ is the two-sample t-statistic to compare the mean of the correlation of two genes with the index, which refers to the position within the rank-ordered miRNA expression from i+1 to j, to the mean of the correlation of the rest of the genes. That is,

$$Z_{ij} = \left\{ \frac{1}{j-i} + \frac{1}{n-j+i} \right\}^{-1/2} \left\{ \frac{S_j - S_i}{j-i} - \frac{S_n - S_j + S_i}{n-j+i} \right\} \tag{5}$$

where $S_i = X_1 + \cdots + X_i, S_j = X_1 + \cdots + X_j (1 \leq i < j \leq n)$ are the partial sums. If the p-values are smaller than a threshold level $\alpha$ (typically 0.01), a change is declared to be statistically significant. The locations of the change-points as the i and j that maximize the test statistic are also estimated by using either Monte Carlo simulations [51] or the approximation approach [52]. After performing the segment clustering approach, only a few groups of samples showing high correlation remain for further analyses. We have a basic assumption here for the ceRNA triplet: the correlation values should be stable across the sliding window approach and thus the correlation values should not be bouncing up and down within a small sample size. Following this assumption, we performed the peak merging step to ensure two nearby peaks were merged into one under certain criteria.

## The *PeakMerging* method

This method is designed to prevent smashed segments resulting from noise. First, two segments are merged when the difference of their correlation values is smaller than a predefined value, and whether the finalized segments are peaks or troughs is defined compared to the baseline. Then, the Fisher transformation is performed to test the difference between two adjacent peaks by comparing their differences against the null hypothesis. Two adjacent peaks are further merged if there is no statistically significant difference between them. Because we presumed it is less likely that two genes are highly correlated to compete for a miRNA at more than two miRNA expression intervals, the triplet was abandoned when more than two peaks occurred in such a relationship. Lastly, candidate ceRNAs are finally selected when their correlation pattern contains a peak with a correlation value over a predefined threshold value (0.3, 0.5, or 0.7).

The final output of the *ceRNAMethod* function provides information on each miRNA, its candidate ceRNA pairs, its peak position within the rank-ordered miRNA expression interval, and the number of samples involved in such ceRNA interactions (S12 Fig).

## Downstream functional analyses

To characterize the biological functions and pathways of identified ceRNA pairs, we implemented the *ceRNAModule* function to generate ceRNA modules from their interaction networks. For the ceRNA modules, the *ceRNAFunction* function is used to perform functional enrichment analysis based on two ontology databases, the Gene Ontology database (GO, http://geneontology.org/) [53] and the Kyoto Encyclopedia of Genes and Genomes Pathway Database (KEGG, https://www.genome.jp/kegg/) [54]. Survival analysis has been widely used in biomedical fields to indicate whether the ceRNAs in the discovered interaction module are associated with the survival of cancer patients. Hence, in ceRNAR, we implemented the *ceRNASurvival* function to perform survival analysis. First, the risk score of each sample is calculated using a multivariate Cox model. All the samples (i.e., different patients) are divided into high or low risk groups based on their median risk scores. The Kaplan-Meier method with a log-rank test is used to test and visualize the difference between the high and low risk groups. Additionally, the *ceRNALocation* function allows users to further visualize the expression level of a specific miRNA when modulated by a specific ceRNA. We also implemented an integrated function (*ceRNAIntegrate*) to combine our results with other state-of-the-art tools, such as SPONGE [20] and JAMI [23], and a validation function (*ceRNAValidate*) based on the miR-Sponge database. The graphical outputs of the above-mentioned functions are presented in S13–S15 Figs.

## Simulation study

In the simulation study, two types of expression data (ceRNA and miRNA) of 100 samples were generated. In each triplet, we presumed 10–40% of samples have correlated genes whereas the rest of the samples do not. The simulated expression profile of 100 samples of miRNA was distributed uniformly between 0 and 1 and was used to sort our samples. We simulated the gene expression profiles of these target gene pairs under two situations. The first is a naïve profile of the correlated genes simply generated from a multivariate normal distribution with a mean value of 0 and a covariance matrix whose entries are 0.9, while the expression distribution of the uncorrelated genes was specified by setting its mean and variance to zero. Another simulation scenario is mimicking a real-data profile (9,835 pan-cancer samples) generated from a multivariate normal distribution with a randomly selected mean value (±2, ±1 and 0), and a covariance matrix whose entries are also randomly selected from 0.3 to 0.9 (correlation level: all), while the expression distribution of the uncorrelated genes was specified by setting its mean (also randomly selected) at ±2, ±1, and 0, and its variance (randomly selected) at 0 to 0.2 (S16 Fig). We also specified three correlation levels (high = 0.8–0.9, mid = 0.5–0.7, and low = 0.3–0.4) and randomly selected values within them to construct the covariance matrix. The normality test for sample distribution of each gene was conducted by the Anderson-Darling method [55].

Five scenarios were considered and are summarized in S1 Table. The first three scenarios were designed for a single peak (i.e., the highest correlation value of the target genes compared to the baseline) occurring at different locations (i.e., different miRNA expression intervals): lower miRNA expression (left, scenario 3), higher miRNA expression (right, scenario 2), and average miRNA expression (center, scenario 1), which represents specific correlation coefficient values existing at different expression levels of miRNA. The fourth scenario was designed for considering the occurrence of two peaks. A null scenario was also compared to test the validity of the other conditions. Two parameters were set under each scenario: window size (10%, 20%, 30%, and 40%) and the threshold at which to define a peak (0.3, 0.5, and 0.7). Each scenario was simulated 1,000 times. We conducted these simulations using either the complete or fast versions of the algorithm, the latter of which reduced computational complexity by

omitting the random walk step. The proportion of correlated samples (10%, 20%, 30%, and 40%) was also analyzed.

## Real data application for tools comparison and validation

In order to compare many aspects, including the computational cost and the quality of the results, of ceRNAR with the other tools (SPONGE [20], CERNIA [21] and JAMI [23]), which similarly used miRNA and paired target gene expression to identify gene-miRNA-gene triplets, we used subsets of the TCGA pan-cancer atlas with different sample sizes and gene/triplet numbers. We ran these tools with default parameter settings in parallel processing with 8 cores with varying sample sizes and numbers of genes.

Regarding real data application, we retrieved two sequencing datasets (TCGA-LUAD and TCGA-LUSC) from TCGA [56] to demonstrate the potential application of our proposed algorithm. These datasets are all retrieved from public domains (GDG data portal: https://portal.gdc.cancer.gov/). The TCGA-LUAD dataset contained 510 samples from lung adenocarcinoma patients, and the TCGA-LUSC dataset contained 476 samples from squamous cell carcinoma patients. To validate the results, the potential ceRNAs identified by our tool were validated experimentally using the miRsponge database [57], and the ceRNAs shared by these two datasets were further analyzed.

## Supporting information

**S1 Fig. Normality assessment through Anderson-Darling test on sample distribution based on TCGA pan-cancer atlas across genes.**
(TIFF)

**S2 Fig. Performance of our proposed method in four scenarios (I to IV) using naïve simulated data with the complete version.** (A) Synthetic data of correlated genes were generated from multivariate normal distribution with a mean value of 0 and a covariance matrix whose entries are 0.6. (B) Synthetic data of correlated genes were generated from multivariate normal distribution with a mean value of 0 and a covariance matrix whose entries are 0.3. The numbers in blue represent the average number of identified ceRNA-miRNA triplets after 100 simulations.
(TIFF)

**S3 Fig. The performance of our proposed method in four scenarios (I to IV) uses simulated data that mimics real-world data distribution with the complete version.** (A) From a covariance matrix with correlation values ranging from 0.3 to 0.9. (B) From a covariance matrix with correlation values ranging from 0.8 to 0.9. (C) From a covariance matrix with correlation values ranging from 0.5 to 0.7. (D) From a covariance matrix with correlation values ranging from 0.3 to 0.4.
(TIFF)

**S4 Fig. Run time comparisons between published tools and our package.** (A) Run time for different sample numbers on a fixed set of genes using real data. (B) Run time for different numbers of genes (i.e., different numbers of triplets) on a fixed number of samples using real data.
(TIFF)

**S5 Fig. ceRNA regulatory network observed in TCGA-LUAD.** The size of each dot represents the number of bridged miRNAs per ceRNA.
(TIFF)

**S6 Fig. The extended analyses of the observed ceRNA pairs that are related to miRNA hsa-let-17e-5p.** (A) The network of overlapping ceRNA pairs in both TCGA datasets. (B) The distribution of miRNA expression at which specific ceRNA interactions occur. The location of each rectangle indicates the miRNA expression value at which particular ceRNA pairs interact with miRNA has-let-17e-5p, and the depth of the color in each rectangle represents the number of samples that have such ceRNA-miRNA interaction. FPKM, fragments per kilobase million.
(TIFF)

**S7 Fig. Experimental validation of real case study based on miRSponge database.** (A) TCGA-LUAD (The Cancer Genome Atlas Lung Adenocarcinoma). (B) TCGA-LUSC (The Cancer Genome Atlas Lung Squamous Cell Carcinoma).
(TIFF)

**S8 Fig. The basic concept of our idea is illustrated graphically.** (A) Lists of curated miRNAs and their targets are verified from nine miRNA target databases based on the AnamiR package and selected by being either experimentally validated or present in over half of the prediction databases. (B) Our algorithm iteratively evaluates whether each mRNA pair is a potential ceRNA event in each miRNA-target list.
(TIFF)

**S9 Fig. A graphic overview illustrates sorting and correlation calculation in the *ceRNApairFiltering* method.** First, samples $[S_1, \ldots S_N]$ based on the expression vector of each miRNA per triplet (genei, miRNAm, genej) are sorted. Second, the window size is defined, and the correlation between gene pairs among samples for the first window is calculated. Next, the correlation between gene pairs among samples for the second window is calculated, and so on. Finally, a new correlation vector per triplet is created.
(TIFF)

**S10 Fig. A graphic overview illustrates the random walk and permutation test in the *ceRNApairFiltering* method.** First, the k correlation values per triplet are ordered according to the average miRNA expression, $miRNA_k = \sum_{n=0}^{w-1} m_{k+n}^{m,i,j}/w$. The score (S) reflects the degree to which the correlation of a gene pair is overrepresented across all values (i.e., high/low/moderate miRNA expression) of the entire set of ranked miRNA expression levels. It is calculated by walking down the ranked value, increasing the score when encountering a specific miRNA expression value with the correlation of a paired gene over 0.3. If the correlation is less than 0.3 (threshold), the score will be negative. The magnitude of the score depends on the number of samples supported by such correlation values within a window. Subsequently, the permuted miRNA expression data are generated 1,000 times to create a null distribution of the S, and then the empirical P-value is calculated accordingly.
(TIFF)

**S11 Fig. The meanings of score S are based on the values of $P_{obey}$ and $P_{violate}$.** (A) Samples with the highest correlation between target genes are enriched at a higher miRNA expression value, supporting that these two target genes have a higher chance to compete with this miRNA. (B) These target gene pairs do not represent a biologically relevant correlation with the competition of this miRNA, suggesting that no ceRNA event occurs in this triplet.
(TIFF)

**S12 Fig. Main tabular output in CSV format.** Five columns are involved: miRNA name, candidate ceRNA pairs, the start and end of each miRNA expression interval, and the number of

segments.
(TIFF)

**S13 Fig. The graphical bubble outputs of the *ceRNAFunction* function are based on over-representation analysis (ORA).** (A) Based on the KEGG database. (B) Based on the 'cellular component' (CC) categories in the gene ontology (GO) database. (C) Based on the 'molecular function' (MF) categories in the GO database. (D) Based on the 'biological process' (BP) categories in the GO database.
(TIFF)

**S14 Fig. The graphical bar outputs of the *ceRNAFunction* function based on over-representation analysis (ORA).** (A) Based on the KEGG database. (B) Based on the gene ontology (GO) database and stratified by three GO levels: cellular component (CC), molecular function (MF), and biological process (BP).
(TIFF)

**S15 Fig. The graphical outputs of the *ceRNAModule*, *ceRNASurvial*, and *ceRNALocation* functions.** (A) Network analysis among candidate ceRNAs for a specific miRNA. (B) Survival analysis for a candidate ceRNA pair targeted by a specific miRNA. (C) A mix of a box plot and bar plot represents the proportion of all candidate ceRNA pairs targeted by a specific miRNA at their corresponding miRNA expression range.
(TIFF)

**S16 Fig. Summary statistics of the gene expression profile of 9,835 samples in TCGA pan-cancer atlas.** (A) The density plots represent the sample distribution of observed z-scores of 10 genes. (B) The density plot and boxplot represent the sample distribution of observed z-scores of 16,323 genes. (C) The pie chart shows the proportion of the Pearson correlation values of 16,323 genes. All the expression values are log-transformed and z-transformed across cancers. The median expression values of samples across cancers are used for summary statistics.
(TIFF)

**S1 Table. A summary of the five scenarios.**
(XLS)

**S2 Table. Performance of the ceRNAR package in five scenarios under the complete version using synthetic data with a fixed correlation value (0.9).**
(XLS)

**S3 Table. Performance of the ceRNAR package in five scenarios under the fast version using synthetic data with a fixed correlation value (0.9).**
(XLS)

**S4 Table. Performance of the ceRNAR package with different proportions of correlated samples in scenario III under the fast version using synthetic data with a fixed correlation value (0.9).**
(XLS)

**S5 Table. Performance of the ceRNAR package with different correlation cutoffs and merging sizes with a fixed correlation value (0.9) and a fixed window size (40).**
(XLS)

**S6 Table. Performance of the ceRNAR package in five scenarios under the complete version using synthetic data with a fixed correlation value (0.6).**
(XLS)

**S7 Table. Performance of the ceRNAR package in five scenarios under the complete version using synthetic data with a fixed correlation value (0.3).**
(XLS)

**S8 Table. Performance of the ceRNAR package in five scenarios under the complete version using synthetic data with randomly selected correlation values from 0.3 to 0.9.**
(XLS)

**S9 Table. Performance of the ceRNAR package using synthetic data with randomly selected correlation values from 0.8 to 0.9.**
(XLS)

**S10 Table. Performance of the ceRNAR package in five scenarios under the complete version using synthetic data with randomly selected correlation values from 0.5 to 0.7.**
(XLS)

**S11 Table. Performance of the ceRNAR package in five scenarios under the complete version using synthetic data with randomly selected correlation values from 0.3 to 0.4.**
(XLS)

**S12 Table. Performance comparison among tools.**
(XLS)

**S13 Table. Running time comparison among tools.**
(XLSX)

**S14 Table. Summary of the application to the TCGA lung cancer datasets.**
(XLS)

**S15 Table. Top bridging miRNAs and their corresponding hub gene among ceRNAs in the two TCGA datasets.**
(DOCX)

**S16 Table. Hub genes among ceRNA triplets in the two TCGA datasets.**
(DOCX)

**S17 Table. Comparisons with the state-of-art tools.**
(DOCX)

## Acknowledgments

We thank Dr. Melissa Stauffer for editing this manuscript and the National Center for High-performance Computing (NCHC) in Taiwan for providing computational and storage resources.

## Author Contributions

**Conceptualization:** Yi-Wen Hsiao, Lin Wang, Tzu-Pin Lu.

**Data curation:** Yi-Wen Hsiao, Lin Wang.

**Formal analysis:** Yi-Wen Hsiao, Lin Wang.

**Funding acquisition:** Tzu-Pin Lu.

**Investigation:** Yi-Wen Hsiao, Lin Wang.

**Methodology:** Yi-Wen Hsiao, Lin Wang, Tzu-Pin Lu.

**Supervision:** Tzu-Pin Lu.

**Validation:** Yi-Wen Hsiao, Lin Wang.

**Visualization:** Yi-Wen Hsiao, Lin Wang.

**Writing – original draft:** Yi-Wen Hsiao, Tzu-Pin Lu.

**Writing – review & editing:** Yi-Wen Hsiao, Tzu-Pin Lu.

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
