## [Decision Letter · Decision Letter 0]

16 Mar 2022

Dear Prof. Lu,

Thank you very much for submitting your manuscript "ceRNAR: an R package for identification and analysis of ceRNA-miRNA triplets" for consideration at PLOS Computational Biology.

As with all papers reviewed by the journal, your manuscript was reviewed by members of the editorial board and by several independent reviewers. In light of the reviews (below this email), we would like to invite the resubmission of a significantly-revised version that takes into account the reviewers' comments.

The experts that have reviewed your manuscript has raised a number of convergent concerns, including major ones about the lack

of a comparison with existing methods and the insufficient description of the algorithmic procedure.

You are invited to review the manuscript taking all revewer's recommendation into account, including a systematic comparison

with established method to demonstrate in objective terms the additional insight provided by ceRNAR.

Upon resubmission, your manuscript will be returned to the three reviewers.

C. Micheletti

We cannot make any decision about publication until we have seen the revised manuscript and your response to the reviewers' comments. Your revised manuscript is also likely to be sent to reviewers for further evaluation.

Sincerely,

Cristian Micheletti

Guest Editor

PLOS Computational Biology

Ilya Ioshikhes

Deputy Editor

PLOS Computational Biology

The experts that have reviewed your manuscript has raised a number of convergent concerns, including major ones about the lack

of a comparison with existing methods and the insufficient description of the algorithmic procedure.

You are invited to review the manuscript taking all revewer's recommendation into account, including a systematic comparison

with established method to demonstrate in objective terms the additional insight provided by ceRNAR.

Upon resubmission, your manuscript will be returned to the three reviewers.

C. Micheletti

Reviewer's Responses to Questions

**Comments to the Authors:**

Reviewer #1: Authors present ceRNAR, an algorithm, written in R, that is able to predict ceRNA-miRNA triplets. Competitive endogenous RNA (ceRNA) are RNA molecules involved in the gene regulation related to several biological and pathological processes. In particular, they form with a miRNA a small interaction network called ceRNA-miRNA triplets, composed of two RNAs and one miRNA. ceRNAR is based on a correlation approach, and moreover it provides some software modules in order to make downstream analysis over the predicted ceRNA networks, including gene enrichment and survival analysis. Experimental tests were carried out using both simultated and real case datasets. The results, in terms of accuracy, sensitivity and specificity, on simulated studies are good.

The paper is clear and well written. English is good. In my opinion, the experimental part of the paper needs to be improved.

Major remarks:

- It is not clear what is the ground truth in your simulation studies. You should better explain that. Moreover, why did you not consider repository of experimentally validate ceRNA interactions collected in databases like miRsponge (http://bio-bigdata.hrbmu.edu.cn/miRSponge/) in order to validate ceRNAR? The simulation study is not enough.

- Authors did not compare ceRNAR with other existing ceRNA predictors. Although in the Introduction they cited several similar tools, no comparison has been done. For that reason, authors shoud add comparison with tools such as SPONGE [1], CERNIA [2], GDCRNATools [3], spongeScan (http://spongescan.rc.ufl.edu/), and specify the added value ceRNAR can provide with regards to web serivices such as miRTissue_ce [4] and Encori (https://starbase.sysu.edu.cn/).

- The output of ceRNAR should be better described, especially the graphic outputs briefly seen in Fig.4

Minor remarks:

Table 1 is reffered to TCGA datasets, not to the simulation dataset. Please fix it

[1] List M, Dehghani Amirabad A, Kostka D, Schulz MH. Large-scale inference of competing endogenous RNA networks with sparse partial correlation. Bioinformatics. 2019

[2] Sardina DS, Alaimo S, Ferro A, Pulvirenti A, Giugno R. A novel computational method for inferring competing endogenous interactions. Brief Bioinforma. 2017

[3] Li R et al. GDCRNATools: an R/Bioconductor package for integrative analysis of lncRNA, miRNA, and mRNA data in GDC. Bioinformatics 34(14):2515–2517, 2018

[4] Fiannaca, A., La Paglia, L, La Rosa, M et al. miRTissue ce: extending miRTissue web service with the analysis of ceRNA-ceRNA interactions. BMC Bioinformatics 21, 199 (2020)

Reviewer #2: The authors propose a computational pipeline to identify ceRNA-miRNA triplets. Specifically, the idea tested here is based on (a) sorting the datapoints based on the miRNA expression level and (b) computing the correlation between expression of genes using the same sorting. This implies a linear relationship between ceRNAs but does not make any assumption on the dependence of the ceRNAs expression level as a function of the miRNA expression level. At the second stage, a segment clustering method is used, which takes into account the ranked miRNA expression levels. Finally, a peak merging is performed to remove false positives. The method is tested on a synthetic dataset and then applied to a dataset taken from the cancer genome atlas.

The discussed approach is interesting since it promises to be much faster than other approaches proposed in the literature. In addition, the authors provide an implementation. However, the manuscript is poorly readable and it is very difficult to appreciate the performance of this method compared with other approaches. I thus think publication is premature at this stage.

What is crucially missing are (a) comparisons of the reported method with other methods, perhaps done on the synthetic dataset, and (b) a fair assessment of the accuracy of the method (PPV and sensitivity) on the real dataset. In the current form, it is difficult to see the advantage of this method with respect to competitors.

Figure 2 reports results for the synthetic dataset. It is not clear from this figure how the critical parameters (windows size and thresholds) can be optimized in absence of a reference result. Indeed, as fairly written in the discussion, the choice of these parameters in real applications is still to be fine tuned.

The comparison with other identified triplets is merely anecdotal, with a quick mention of two previously reported triplets, but there is no systematic analysis performed. In addition, the authors should compare their method with alternative methods on their synthetic datasets, providing a quantitative assessment on the performance difference (both in terms of computational cost and quality of the results).

Finally, the text should be better proof read. There are a number of incorrect sentences that makes it difficult to read. E.g., "there is still needed for a tool" (abstract) "simulations of different parameters" (line 134; perhaps "simulations with different parameters"?). Some acronyms are defined after they are use and also makes the paper difficult to read.

Reviewer #3: With ceRNAR, Hsiao et al. present a new R package for inferring competing endogenous RNA triplets from paired gene and miRNA expression data. In contrast to prior approaches that leverage conditional mutual information or partial correlation to assess the mutual influence that ceRNAs exert on each other via miRNA binding competition, ceRNAR considers a direct correlation approach for which miRNAs with similar expression are grouped in windows. The likely intention behind this is to fix the miRNA expression to then investigate the competition in absence of miRNA changes. I have the following comments:

#Major:

- The basic intuition of the method is not explained in sufficient detail. Only the sentence "we consider gene expression and known interactions among miRNA-gene pairs at a specific miRNA expression level" point to this but does not explain the rationale behind this approach. It is left to the reader to reverse engineer the underlying hypothesis from the method section.

- While the idea is certainly interesting, I'm not convinced that fixing a miRNA expression level works the way the authors expect. If the idea is to remove the influence of miRNAs one has to consider that miRNA expression changes are dynamic. Several scenarios are plausible for observing a given miRNA expression level. For instance, the miRNA might have been upregulated together with a ceRNA which then compensates for the extra miRNA copies via miRNA binding / sponging. The authors should describe in detail in which scenarios their method is likely to be successful and where it is not.

- The manuscript describes two hypotheses (i) miRNA tends to be sequestered by the miRNA with the higher expression level and (ii) miRNA has a greater affinity for the mRNA with more MRE. It should be highlighted that likely both effects are relevant and mixed.

- The manuscript does not describe the state of the art sufficiently. Several tools such as CUPID / JAMI or SPONGE are missing here.

- Furthermore, a performance comparison of ceRNAR to these tools which infer ceRNA-miRNA triplets is missing.

- Most ceRNA pairs share more than one miRNA and should ideally be considered together as e.g. in the SPONGE method. This is not feasible here since we can sort only for one miRNA but the limitation should be discussed.

- The method descriptions are not easy enough to understand, e.g. in "The ceRNApair Filtering method" it is not explaind what is meant by "window", i.e window of miRNA expression values in the sorted expression vector.

- The whole intuition behind Pobey and Pviolate is lost to me. This makes it hard to understand what the method aims to show here. Please revise this section.

- Similarly, I fail to understand the motivation for the segment clustering and peak merging. What peaks are we talking about?

- Regarding the simulation: why is the mean and variance of uncorrelated genes set to zero? This would mean that no uncorrelated genes are expressed.

- The simulation setting is way too simplistic. Here, the manuscript only assumes a correlation of 0.9 between ceRNA pairs but in reality this can take any value. More realistic would be to set an random correlation of 0.3-0.9 here for each pair. Realistically, gene pairs that are not ceRNAs will also have a meaningful positive correlation which is why typically conditional mutual information or partial correlation needs to be used to determine the miRNA contribution in the triplet. I don't think the simulation results are hence reflecting the performance on real-world data. A more realistic simulation would be needed for a fair method comparison though (see my comment above).

- I could not find any open source license in the github repository. Please add one.

#Minor:

- The manuscript states that "the partial correlation approach considers the expression of the shared miRNAs of two ceRNAs when calculating their correlation and ignores the interaction strength of them". I disagree here. Sensitivity correlation which is computed as the corrlation of two ceRNAs minus the partial correlation accounting for the miRNA which is referenced here as [14] certainly captures the effect size but albeit not the significance of an interaction. A measure of significance for sensitivity correlation was later suggested in 10.1093/bioinformatics/btz314.

- Figure 4: text is too small to read anything

**Have the authors made all data and (if applicable) computational code underlying the findings in their manuscript fully available?**

Reviewer #1: Yes

Reviewer #2: Yes

Reviewer #3: Yes

PLOS authors have the option to publish the peer review history of their article (what does this mean?). If published, this will include your full peer review and any attached files.

Reviewer #1: No

Reviewer #2: No

Reviewer #3: **Yes: **Markus List
---

## [Decision Letter · Decision Letter 1]

7 Aug 2022

Dear Prof. Lu,

Thank you very much for submitting your manuscript "ceRNAR: an R package for identification and analysis of ceRNA-miRNA triplets" for consideration at PLOS Computational Biology. As with all papers reviewed by the journal, your manuscript was reviewed by members of the editorial board and by several independent reviewers. The reviewers appreciated the attention to an important topic. Based on the reviews, we are likely to accept this manuscript for publication, providing that you modify the manuscript according to the review recommendations.

Dear Prof. Tzu-Pin Lu,

The consulted experts have evaluated favourably the revised version of your manuscript and I am ready to recommend its acceptance into Plos. Comput. Biol.

Towards that I wish to ask you to implement the minor changes that reviewer 3 has constructively recommended to improve the clarity and completeness of the manuscript.

The recommended textual changes are very specific and I trust that they can be implemented quickly and with modest effort.

Yours sincerely,

C. Micheletti

Sincerely,

Cristian Micheletti

Guest Editor

PLOS Computational Biology

Ilya Ioshikhes

Deputy Editor

PLOS Computational Biology

[LINK]

Dear Prof. Tzu-Pin Lu,

The consulted experts have evaluated favourably the revised version of your manuscript and I am ready to recommend its acceptance into Plos. Comput. Biol.

Towards that I wish to ask you to implement the minor changes that reviewer 3 has constructively recommended to improve the clarity and completeness of the manuscript.

The recommended textual changes are very specific and I trust that they can be implemented quickly and with modest effort.

Yours sincerely,

C. Micheletti

Reviewer's Responses to Questions

**Comments to the Authors:**

Reviewer #1: Authors succesfully answered my remarks.

Reviewer #2: The manuscript is significantly improved. In my opinion it is suitable for publication

Reviewer #3: The authors have done an admirable job improving the manuscript and in responding to the reviewer requests. Many of my previous questions have been answered. The underlying hypotheses are now much more clearly defined and the methods are better explained, especially through the figures that have been added. I would nevertheless recommend to include some of the explanations from the reviewer response into the manuscript, e.g. where the authors explained the intuition behind Pobey and Pviolate as well as the segment clustering since I believe this will make it much easier for readers to understand. The comparison with related tools that was now included offers a good overview of existing approaches. A few comments remain to be addressed:

- Figure S7: the caption does not explain what the legend labels mean. What is the meaning of no_validated, validated_both, validated_each?

- In a response to reviewer 1 ("Authors did not compare ceRNAR with other existing ceRNA predictors"), the authors claim that all five tested methods are correlation based. Just to clarify this I want to point out that mutual information and correlation are two distinct concepts, meaning that CUPID and JAMI which employ conditional mutual information can not be considered correlation-based methods.

- Table S17: While I really appreciate the comparison table for the different tools, I'd like to point out some issues. Some of the tested tools have no built-in "miRNA-targets data sources", thus it appears unsuitable to list specific data sources here. One should specify that the tools are flexible. For example saying "user-provided, e.g. miRecords…". The table probably needs a caption since some rows are unclear. For example, what is meant by the row "TCGA expression". Most tools are agnostic to the source of the expression data. Similarly, it is not fully clear what is meant by "case-only".

- The simulation experiment that is meant to show the performance of the various too-ls is based on a pre-built covariance matrix. This approach leads to linear relationships, giving correlation-based methods a decisive advantage here. The poor performance of JAMI can be attributed to this, as no non-linear relationships can be discovered here. However, simulating non-linear relationships is not realistically feasible and the authors should just discuss openly that this evaluation is unfair for JAMI.

- Figure 3: The caption does not explain the different scenarios.

**Have the authors made all data and (if applicable) computational code underlying the findings in their manuscript fully available?**

Reviewer #1: Yes

Reviewer #2: Yes

Reviewer #3: Yes

PLOS authors have the option to publish the peer review history of their article (what does this mean?). If published, this will include your full peer review and any attached files.

Reviewer #1: No

Reviewer #2: No

Reviewer #3: **Yes: **Markus List

Figure Files:

Data Requirements:

Reproducibility:

References:

---

## [Editor Report · Decision Letter 2]

18 Aug 2022

Dear Prof. Lu,

We are pleased to inform you that your manuscript 'ceRNAR: an R package for identification and analysis of ceRNA-miRNA triplets' has been provisionally accepted for publication in PLOS Computational Biology.

Best regards,

Cristian Micheletti

Guest Editor

PLOS Computational Biology

Ilya Ioshikhes

Section Editor

PLOS Computational Biology

The authors have addressed the additional minor changes recommended by one of the reviewer.

---

## [Editor Report · Acceptance letter]

31 Aug 2022

PCOMPBIOL-D-22-00061R2 

ceRNAR: an R package for identification and analysis of ceRNA-miRNA triplets

Dear Dr Lu,

I am pleased to inform you that your manuscript has been formally accepted for publication in PLOS Computational Biology. Your manuscript is now with our production department and you will be notified of the publication date in due course.

With kind regards,

Zsofia Freund
